# Creative Arts Therapies as Temporary Home for Refugees: Insights from Literature and Practice

**DOI:** 10.3390/bs7040069

**Published:** 2017-10-17

**Authors:** Rebekka Dieterich-Hartwell, Sabine C. Koch

**Affiliations:** 1Department of Creative Arts Therapies, School of Nursing and Health Professions, Drexel University, 1601 Cherry St., Philadelphia, PA 19102, USA; 2Research Institute of Creative Arts Therapies (RIArT), Alanus University, Alfter 53347, Germany; 3Therapeutic Sciences, Dance Movement Therapy, SRH University Heidelberg, Heidelberg 69123, Germany

**Keywords:** refugees, homesickness, home, creative arts therapies, enactive transitional space, embodied aesthetics

## Abstract

One of the frequently overlooked psychosocial problems of refugees is the phenomenon of homesickness. Being forced into exile and unable to return home may cause natural feelings of nostalgia but may also result in emotional, cognitive, behavioral and physical adversities. According to the literature, the creative arts therapies with their attention to preverbal language—music, imagery, dance, role play, and movement—are able to reach individuals through the senses and promote successive integration, which can lead to transformation and therapeutic change. These forms of therapy can be a temporary home for refugees in the acculturation process, by serving as a safe and enactive transitional space. More specifically, working with dance and movement can foster the experience of the body as a home and thus provide a safe starting place, from which to regulate arousal, increase interoception, and symbolize trauma- and resource-related processes. Hearing, playing, and singing music from the home culture may assist individuals in maintaining their cultural and personal individuality. Creating drawings, paintings, or sculpturing around the topics of houses and environments from the past can help refugees to retain their identity through art, creating safe spaces for the future helps to look ahead, retain resources, and regain control. This article provides a literature review related to home and homesickness, and the role the arts therapies can play for refugees in transition. It further reports selected interview data on adverse life events and burdens in the host country from a German study. We propose that the creative arts therapies are not only a container that offers a temporary home, but can also serve as a bridge that gently guides refugees to a stepwise integration in the host country. Several clinical and research examples are presented suggesting that the support and affirmation through the creative arts can strengthen individuals in their process of moving from an old to a new environment.

“Home is where I can grow” (“Heimat ist, wo ich wachsen kann”) Title of a theatre play by female refugees from Iran

Of all the transitions experienced in life—graduating from school, starting a new job, getting married, losing a grandparent, and having a child—moving away from home is filled with particular challenges and meaning. A new environment with new sounds, colors, smells, and visceral sensations takes time to adjust to. As time passes, memories remain. The melancholy song of a local bird, the familiar path to a friend’s house, the whiff of a traditional dish, the warm hug of a loved one from the past, all these sensual memoirs can create nostalgia or homesickness, a longing for a place and time from long before. This longing is especially trying when what is perceived as home is no longer accessible. For refugees who cannot return to their place of origin due to war or other hardships, there is an added complicated factor: home may not only be associated with longing but also with fear and other negative emotions. While the new environment offers safety and, possibly, better opportunities, it is unfamiliar and different, thus leaving a person with feelings of ambiguity and doubt.

## 1. Home and Homesickness for Refugees

What is home? According to Mallett [1], home can be abstracted as a multidimensional concept that includes definitions from disciplines such as sociology, anthropology, psychology, geography, architecture, and philosophy. Home can be a physical dwelling, a country, or a birth place, but it is not necessarily a singular fixed place or state of being [2]. Instead, it can be multiple places, family, other relationships, a feeling, or a practice [1]. The feeling of being home may at times be unattainable as there is a human tendency to seek for the ideal home, a sentimental search for a lost time and space [1,3]. Every person has a unique definition of what home means and how it feels, but frequently this phenomenon is only perceived when away [4], as journeys establish the threshold and borders of what constitutes home [1,5]. Metaphorically, home can have a range of meanings as well, from being at ease and comfortable with something (“make yourself at home”) to the root of a matter (“it struck home”) [6].

Homesickness or nostalgia is a condition that was first described in relation to its pathological implications in 1688 by a Swiss doctoral medical candidate named Johannes Hofer [7]. Although it has been known for centuries, particularly in the American literature, little attention has been paid to this phenomenon [8]. However, according to Frigessi-Castelnuovo and Risso [7], facing a foreign society with different customs, norms, values, behaviors, rhythms, and relationships to time and space is actually a strike at one’s cultural identity and cause for nostalgia in immigrants. Other determining factors are the degree to which an individual willingly entered exile, the nature of departure (e.g., sudden and without preparation or possibility to say goodbye) and the absence or presence of contact and work in the new environment [4]. While homesickness is a normal occurrence, it may become pathological when an individual is unable to cope adequately with the emerging feelings [8]. Depression, loss of control, obsessive thoughts about home, and apathy are some of the emotional, cognitive, and behavioral symptoms noted [4,9]. Physically, individuals may experience gastric and intestinal problems, sleep disturbances, appetite loss, headache, fever, and aches and pains [4,10]. Homesick individuals have little interest in the new environment, which makes adaptation and settling virtually impossible [6]. For some individuals, the only cure is to return back home where recovery can be speedy and significant [8]. 

According to Taylor [11], refugees are “in the crudest way defined by the loss of home” (p. 130). Whether they are pathologically homesick or able to cope with the feelings, all refugees share a “deep sense of nostalgic yearning for restoring their very specific loss” [6] (p. 15). Koch’s [4] study, mentioned below, found homesickness to be the most important aspect of experienced stress in the host country among the refugees she interviewed. Unique about the refugee predicament is the complex and pervasive, yet fundamental and primary nature of their loss that results in a nostalgic disorientation and an incomprehensible gap [6]. This can be conceptualized as a state of limbo [12], a liminal space in which the individual is no longer at home but does not yet feel at home in the new environment. Liminality may be experienced with discomfort and uncertainty, yet this “in-between space” carries a significant potential for transformation and change [13].

## 2. Aesthetics: Arts as Both Sensual and Transcendental

Before we communicate verbally, we create sounds, move our bodies through space, and fill paper, walls, or furniture with scribbles and figures. These preverbal forms of communication and expression remain lifelong significant sources of knowledge that embrace imagination, symbolism, and the senses [14]. Aesthetics, from the Greek word aisthētikós (“sense perception”) is a branch of philosophy that examines art, beauty, and taste along with the creation and appreciation of beauty. Aesthetics as an epistemology is a way of knowing that differs from cognitive, purely rational, and analytic procedures and instead creates a sensory, kinesthetic, and imaginary understanding [15]. The German philosopher Heidegger proposed that art is truth [16]. He gave the example of Van Gogh’s depiction of a pair of peasant’s shoes with the well-worn insides and the leather that implies the dampness and richness of the earth and suggested that this artistic portrayal showed what shoes, in truth, were. Gadamer [17] went a step further and postulated that art induces an elevated or transcendental state of being. He also affirmed that art and symbols transport us through time, allowing us to travel back in history and bringing past experiences into the here and now, as a holder of memory and history [17]. In other words, images through visual art, music, dance and movement all precede verbal language and can retrieve one’s personal past narrative. Thus, the creative arts are in the present, but also build bridges to the past and to other ways of knowing and being. 

## 3. Creative Arts Therapies and Refugees: Aesthetic Experience Can Be a Shelter

The creative arts therapies (CATs), including music therapy, art therapy, drama therapy, and dance/movement therapy, provide a symbolic language through their various art forms that seeks to access “unacknowledged feelings and [provides] a means of integrating them creatively into the personality, enabling therapeutic change to take place” [18]. According to Dokter [19], the CATs can be utilized to “support the maintaining of a cultural identity, especially in situations where some of that identity is lost or is in conflict with the dominant culture” (p. 16). Refugees may also be able to externalize their symbolized trauma through visual art, music, dance, or drama before they can access these verbally and, upon integrating, experience strength and positivity [20]. The arts function as a shelter, the active involvement with them functions as taking ownership of constructing that temporary shelter. Aesthetic pleasure can be experienced like a protective cloak, shielding oneself from the aversive environmental conditions, bringing back a feeling of wholeness. Active creation of such aesthetic pleasure can be the means of experiencing resources, self-efficacy and resilience [21]. These assumed active factors need empirical testing.

## 4. The Creative Arts Therapies as a Temporary Home

Throughout the literature, there are implicit suggestions that art, music, drama, and dance or movement in therapy can offer a safe place or temporary home that assists in restoration and integration. Furthermore, all of the arts therapies support storytelling, the narration of one’s story (or parts thereof), and the expression of that “what needs to be expressed”. If a translator is included into therapy, the therapeutic relation becomes a therapeutic triangle with trust needing to be established between all of the persons. The presence of a translator can, thus, sometimes be a challenge to the therapeutic process. However, when successful, it can facilitate integration of past, present and future themes/aspects with even more than one person of trust. Figure 1 depicts the relationship between home and the creative arts therapies.

### 4.1. Dance/Movement Therapy

Dance/movement therapy (DMT) is based on the premise that body, mind, and spirit are interconnected and that the body reflects unconscious processes [22]. Embodiment, a DMT related concept that integrates physical, phenomenological, kinesthetic, and movement-based perspectives of an individual has been suggested to entail three levels: the embodied self (union of mind and body), the enactive self (living system in environment), and the extended self (embodied self, reaching into cultural environment) [23]. According to Koch and Fuchs [23], the body “is the unifying base of the constant first-person perspective that we carry with us” (p. 278). The concept of the self, in turn, is closely linked to home [24]. The physical body, too, has been called home, a dwelling for life [25], and an indwelling [26]. Meeks [24] suggested that working with the metaphor body as home in the context of DMT could promote individuals’ sense of security, control, and comfort, encouraging healthy attachment, authenticity, and an improved body image. Specifically, for homeless and displaced populations, the exploration of *body as a home* can assist in creating “a mobile sanctuary in the body, or help them to feel the beneficial qualities of home within” ([24] (p. 80); [27]).

### 4.2. Drama Therapy

Drama therapy facilitates the embodied expression of internal conflicts and the rehearsing of alternative choices through role play, improvisation, and storytelling [28,29]. With roles and masks providing distance, participants can act out tensions in a safe environment, try on different identities and play with metaphors [29]. Roles offer an opportunity to engage with oneself from varying distances (e.g., a human figure is closer to the self than an animal figure; Orkibi, 2017 [30]), which helps clients to create their own save space and regulate the distance as needed. According to Scott-Danter [31], “the story and issues are recognizable and yet theater distances them as fiction” (p. 108). When feelings are physically represented through movement, statues, and sculptures, language can be transcended [32].

### 4.3. Music Therapy

Music therapy (MT) rests on the foundational principle that music is unique to being human [33] and plays a role in the evoking of emotions [34]. While music is not an end in itself, it is used as a means to an end [35]. In this modality, active and receptive experiences are offered to promote mental and physical health and reach individualized goals [36]. Although our preferences for music are individual, they are grounded in our culture [35]. In community music therapy, a branch of MT that is particularly flexible and can be adapted to different cultures [37], therapists are called to be knowledgeable about their clients’ musical traditions. As individuals play, sing, and listen to songs from their home cultures, they reenter a specific emotional state that helps them connect to their inner resources for growth, maintain their cultural and individual personality [38], and yet be more fully present in the new environment [39]. In this way, they can create a temporary home that feels safe and fosters integration [40]. 

### 4.4. Art Therapy 

The underpinning of art therapy is the healing and life endorsing nature of the creative process of art making [41]. It is believed that all people have the capacity to express themselves creatively with the process and not the product being the most essential byproduct [42]. According to Rubin [43], creating art and perceiving art are both ways of making and finding meaning. An increasing demand for culturally sensitive art therapy has established guidelines that include valuing of diversity, multicultural competence, and promotion of empowerment [44]. When individuals create art, memories are released. The physical act of the art process, which includes hand and body motions, further offers moments of regained power and aesthetic experience [45]. As individuals depict their old houses, homes, loved ones, aspects of themselves, and their stories through visual media, their identities can be strengthened [45] and they can experience a sense of momentary home, stability, and remembrance [46].

### 4.5. The Container and the Bridge 

In the descriptions above, we learned that the creative arts therapies (CATs) can serve as a temporary home. This expression has two connotations. As the home, it may represent a safe haven, a place to be oneself, a container that holds and keeps. The temporary aspect on the other hand speaks to the experience of change, transformation, and bridging the old and the new. Figure 2 reflects the characteristics of the CATs, the container and the bridge, visually. 

Badakhshan [47] described this phenomenon as follows: “The artistic expression as a form of nonverbal communication facilitates a sense of nonverbal community. While the cultural origins can be authentically relived through the art process, the new culture is experienced and approached. This state fosters a connection between both cultures and leaves room for an intercultural space” (p. 3). Based on examples from the literature, we propose that the CATs may support the gradual integration of refugees by serving initially as a container and secondly as a bridge to the new culture. 

## 5. Clinical and Research Examples 

### 5.1. State of Limbo

Callaghan [12], who conducted DMT groups with male refugees from African and Asian countries in the U.K., noticed that commonly they felt that their bodies were not part of themselves. Similarly, Badakhshan [47] in his music therapy project with mainly Iranian refugees in Germany found that the participants were grappling with their identities and the sadness about being far from home. Art therapist Heriniaina [48] described a loss of reference and a presence of “patchwork identity” (p. 92), an arbitrarily pieced together sense of self, in the refugee clients she worked with. 

### 5.2. Creative Arts Therapies as Container

Through movement and dance, this state of limbo can be “accompanied” and given definition. Callaghan [12] illustrated how in the beginning phase of her refugee group, a form was created that helped establish a structure and reduce uncertainty. This type of container is also readily found in art, drama, and music therapy, where art materials, stories, and instruments provide a natural framework. Familiarity and safety are important, particularly in the beginning, as individuals “test the waters” of yet another new experience. Furlager (personal communication, 8 February 2017) stated that “depending on the culture, DMT can be a way for refugees to do what they already know to do…many African countries have a culture of singing and dancing together in a circle and in this way the refugees can share their special cultural identity…they can access a language that is familiar to them.” The refugees in Callaghan’s [12] group, for example, spread out a big cloth in the style of a Middle Eastern table, and gathered around it, recalling meals and celebrations in their home countries. 

Sajnani and Nadeau [49], who worked with refugee females in the context of the Creating Safer Spaces program in Montreal, Canada, found that developing a forum theater resulted in a greater sense of togetherness and “collective determination” (p. 51). Refugees can find their own language in drama therapy [50]. As they tell their own tale and explore the setting of their stories, they can begin to share central themes with others. 

Singing or playing songs from the culture of origin may not only strengthen individuals’ sense of self and maintain their personal and cultural individuality, but can also be reassuring. In a recent study of Badakhshan [47] with refugees in Heidelberg, all interviewees preferred to listen to the music from their home countries, and some did not listen to the music from the host country at all. Most described an emotional relationship to their music and that they used it to express or regulate their emotions for other similar non-musical goals [47]. Badakhshan concluded that the music from the home country is of central relevance. The therapist needs sufficient knowledge about the musical culture of his/her clients to offer their music in an appropriate way in therapy [47] (p. 44).

Shapiro [38] depicted how the very act of men singing in their foreign tongue appeared to make the statement “I am here” and confirmed their being seen and heard [51]. The idea of music therapy may be new and strange for refugees, but music making is frequently a natural part of the communal life in their society of origin [52]. Even when the clients in a group do not share a culture, communicating their own backgrounds through music and song may lead to a level of trust and solidarity, a new identity within their new community that can feel like home [52]. 

A sense of container and organization is generated when clients cast clay models or create Mandalas [48]. Particularly, when individuals are weary and lack effort, activities that are clearly defined, and leave no room for uncertainty, can help them externalize and produce something aesthetically pleasing in a relaxed atmosphere [48]—an object, they can relate to and that provides the opportunity for dialogue and eventually for a good-bye or some other sort of closure [53]. Heriniaina [48] found that her refugee clients frequently drew, painted, or constructed houses and landscapes that were reminiscent of their old homes. For example, one client built a bird house that for him signified escape, freedom, and home [48]. Such depictions are a way to keep alive parts of their identities, create a safe representation of their homes, and symbolically attempt an expansion. 

As they learn to master musical instruments and art materials and connect with their bodies and their own stories, refugees can regain a sense of power and control which may open up venues for transformation and change. Zharinova-Anderson [39] sums up this phenomenon for music therapy, but dance, drama, and art therapy can certainly be utilized in a similar manner: “Music therapy [helps] clients to contact their inner resources for growth and by using their own cultural music it allows them to be more present in their new culture as whole human beings. In changing the way refugees feel about themselves and are perceived by their new communities, music can become a ‘force for change’” (p. 245).

### 5.3. Creative Arts Therapies as Bridge

Over time, aspects of the new environment are slowly and organically introduced in the art modalities. Callaghan [12] spoke of the “revolving door”, a time of transition, when she noticed that her clients were weaving through the group. This phase is an important time of experimenting and practicing self-expression. The therapist needs to play the role of a witness and “provide a mirror for the individuals to look into themselves, to facilitate the process of change and create the necessary environment for the individuals to explore themselves” [54] (p. 188). Clients are encouraged to find their own dance and slowly integrate all the different aspects of their world—the old, the new, the inside, and the outside. Harris [55] in his work with South Sudanese refugee took on a role as facilitator, offering communal traditional Dinka dancing and drumming that was based on their cultural rituals. He noticed that over time, their collective resilience was increased as well as their capacity to acculturate and integrate into the host country. The women began to drum (traditionally only males drum) and ranges of individual expression became more common with roles shifting subtly [55]. 

A playback theater program for refugee high school students in Montreal, Canada allowed the participants to share group stories, construct meaning, and ultimately create a “bridge between the past and the present” [56] (p. 539). According to Landis [32], making meaning from past stories helps individuals find new roles in the new home country [32].

Through communal music making, refugees can overcome isolation and step into a process of acceptance and integration into the new life [52]. In this process, natural inclinations and tendencies can be utilized. For example, Baker and Day [57] described the affinity of many Sudanese youth towards Western hip-hop artists. With inborn improvisational skills often noted in Sudanese youth, the musical form of rap is believed to function as a “vehicle for both the verbal processing of feelings and the internalizing of new societal values” [57] (p. 99) and may thus serve as a bridge from the old to the new environment. A very different example was presented by Amir [58] who worked with native Israelis, some of them Holocaust survivors, who were dealing with bereavement. She portrayed how singing Israeli folksongs helped the participants remember and increase their emotional expression. The ritualistic nature of the experience created safety, yet the choices given encouraged independence and inner freedom. One of the songs became a mutual symbol of the future, of courage, and of change [58].

When refugees have processed their feelings of loss through their art work, they may be ready to engage in a technique entitled “The Collage of Your Life” [59], in which they paste and glue pictures, words, or other visualizations that represent different aspects of themselves. Wong-Valle [59] who worked with Puerto Rican immigrants found that this activity resulted in a convergence of images that could guide them to combine and integrate their bi-cultural realities. Visualizing parts of their old selves and parts of their developments in the new environment together allowed them to bridge the gap. Similarly, Fitzpatrick [46] illustrated that the process of art making gave Bosnian refugees to Australia an opportunity to experience themselves in a new way, which boosted their range of emotional choices and promoted identity integration. One of the strengths of visual art is that it is transportable and lasting. When refugees are able to create “beautiful” and aesthetically stimulating pieces, such as paintings, mobiles, and sculptures for their new housing environments, they tend to invest themselves in the here and now of their settling process [48]. In other words, they have “arrived” just a little bit more.

### 5.4. Homesickness: Selected Results 

Empirical data on homesickness are reported in Koch [4], in a mixed methods study that employed semi-structured interviews with 20 mixed-nation refugees of insecure status in Heidelberg. The sample consisted of 15 men and five women, with a mean age of 23 years (*SD* = 4.1; range: 17–30) who were in Germany for between 0–5 years. They came from Iran, Iraq, Afghanistan, and Eritrea, and were 15 asylum seekers and five “Geduldete (GFK)”, with a temporary residence allowance following the Geneva Refugee Convention (GFK), thus, all 20 of insecure status. Koch [4] asked among other questions about: (a) the life events of the person (what have you experienced in your home country and on the flight?); and (b) the biggest stressor in their lives in the host country (from what do you suffer most here in Germany)? Questions were semi-structured with items taken from the Life-Event-Questionnaire [60], plus the additional option to add other life events (about half of the events were added by the refugees themselves). Results are provided in Table 1 and Table 2.

Homesickness was the most often named stressor for this sample of refugees in Germany. Even though we have not assessed the degree or the pathological quality of homesickness in this study, it became clear that the psychological burden from it was serious. In one refugee, it was so profound that he developed a conversion disorder: he was not able to walk anymore and needed to be in psychiatric treatment. He attributed this solely to his homesickness and felt that the only cure was to return back home [4].

## 6. Obstacles to the Process of Successful Transitioning 

The transition from one culture to the other is not always successful.

*Unsuccessful patterns*: The process of transitioning from container to bridge can also become a vicious circle as described in a single case study in the dissertation of Widdascheck [61]. Unsuccessful transitioning can, among other aspects, take the forms of: (a) a *constant in limbo state* [61]; (b) a *turning backward*: falling back and possible extremization of values from the home culture, often with a strong experience of the losses and a development of an enduring attitude or increasing homesickness; or (c) a *turning forward*: total abandoning of the old culture with the danger of a loss of the roots and anchor in one’s own culture; often also loss of social networks. Impaired health states can result from all three described patterns [20].

*Gender aspects*: Women do often adapt more readily to a new culture. However, because of different values of their home countries, they are then sometimes “stopped” or “slowed down” in their acculturation process by their families. For many refugees, this creates a new form of gender tension, and new challenges to successful gender value transition and social role adaptation [62]. According to Binder and Tošic [62], female refugees from patriarchal social structures are particularly vulnerable and are “rendered passive in a double sense, as refugees and as women” (p. 622).

*(Sub-)Cultural aspects*: The degree to which persons are culturally open or religious might also have effects on the success of cultural transitions. For example, Iversen, Sweaass, and Morken [63] learned that Asian refugees were more likely to want to learn the host language than their African counterparts. A recent study on the psychological health of Russian Jewish Immigrants in Austria found that the acculturation attitude influenced the prevalence of depression in this migrant population, with those being more open to the host culture suffering less depression [64].

## 7. Conclusions

The CATs seem to have the possibility to reach people regardless of their cultural backgrounds. We assume that this characteristic is partly due to the enactive aesthetic experience (with active factors such as experience of beauty, creation, generativity, authentic expression, being moved; Koch 2017 [21]) and can be particularly valuable when it comes to serving refugees in their acculturation process. For those who struggle in a state of limbo and feel like the old is no longer accessible, yet the new seems so far away, the creative arts may provide a temporary home. In the hands of a skilled therapist, this place of art-making through visual art, music, drama or dance and movement, can not only represent a safe container (safe space) that encourages authenticity and familiarity – a journey on which clients can integrate different pieces of their identity through self-expression and aesthetic creation – but also build a bridge to the new environment (enactive transitional space), so that the host country can become a home country. 

## Figures and Tables

**Figure 1 behavsci-07-00069-f001:**
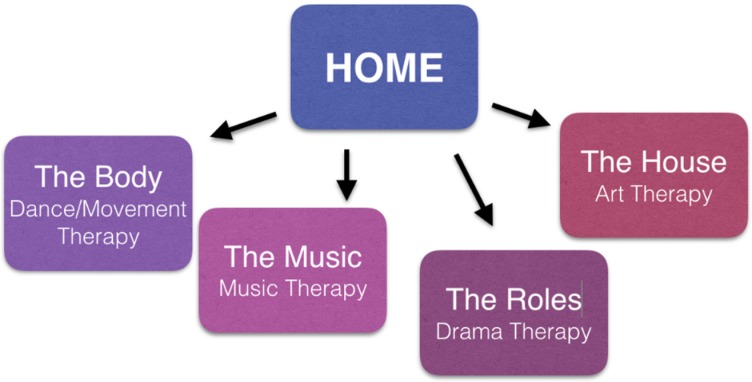
Home and the Creative Arts Therapies.

**Figure 2 behavsci-07-00069-f002:**
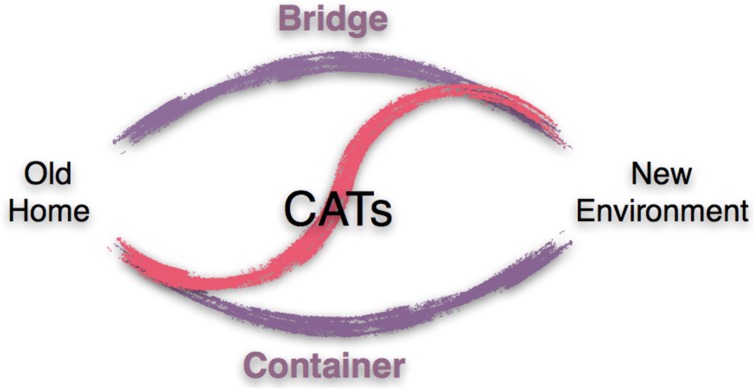
Creative Arts Therapies (CATs) as Container and Bridge.

**Table 1 behavsci-07-00069-t001:** Life events in home country and on the flight (*N* = 20).

Life Events before Arrival in Germany	# of Persons Affected
Torture	3 persons
Starvation	5 persons
Danger of Own Death	7 persons
Loss of Work	9 persons
Imprisonment	10 persons
Death of Close Relatives	10 persons
Social Decline	11 persons
Discrimination	13 persons
War	13 persons
Death of Friends	14 persons
Hiding and Illegality	15 persons
Persecution of Friends and Relatives	16 persons
Separation from Family and Friends	17 persons

Note. Multiple answers were possible; the sample consisted of 15 men, 5 women; mean age 23 years (*SD* = 4.1; *range*: 17–30); in Germany 0–5 years; from Iran, Iraq, Afghanistan, and Eritrea, all of insecure status; data from semi-structured interviews of Koch [4].

**Table 2 behavsci-07-00069-t002:** Most serious stressors in Germany (N = 20).

Stressors in Germany	# of Persons Affected
Loneliness	2 persons
Language Difficulties	3 persons
Hostility toward Strangers	4 persons
Poor Living Conditions	4 persons
Loss of Freedom to go anywhere one wants	8 persons
Experiences of Persecution and Flight	8 persons
Lack of Work	9 persons
No Passport	10 persons
Insecurity of Status (in Asylum Process)	10 persons
Dependency on Social Welfare	12 persons
Insecure Residential Status	12 persons
Homesickness	16 persons

Note. Multiple answers were possible; the sample consisted of 15 men, 5 women; mean age 23 years (*SD* = 4.1; *range*: 17–30); in Germany 0–5 years; from Iran, Iraq, Afghanistan, and Eritrea, all of insecure status; data from semi-structured interviews of Koch [4].

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
