# Peer review of "Creative Arts Therapies as Temporary Home for Refugees: Insights from Literature and Practice"

_behavsci, 2017, doi:10.3390/bs7040069_

Round 1

Reviewer 1 Report

Thank you for this well-written assertion that the arts therapies may offer a form of shelter or home for those who have been displaced. I think it could be a wonderful resource but also have a few suggestions as to how to strengthen it. At present, neither the abstract not introduction indicate that your assertion about arts therapies as home, container and/or bridge will be derived from a literature review. Setting this up as your methodological approach would significantly strengthen your primary assertion. Secondly, the claims to be about the creative arts therapies but, for the most part excludes drama therapy and the reasons for this are unclear. A simple search would reveal Ditty Doktor's work in this area, Rousseau et al's study into the effectiveness of playback theatre with adolescent acculturation, and Sajnani and Nadeau's examination of the use of Theatre of the Oppressed with refugee women - and these are just a few examples. There is also a danger of downplaying the importance of cultural literacy and language, i.e. through trained interpreters, when working with vulnerable populations. Finally, the section on obstacles to successful transitioning seems tacked on at the end when it may serve as a useful way of beginning the article which offers a unifying metaphor (home) derived from clinical examples of how the arts therapies support the acculturation process. 

Author Response

Dear Reviewer,

please see the attached document for our responses.

Thank you!

Reviewer 2 Report

This paper presents an overview of research to date pertaining to the field (CAT) and a synopsis of research undertaken by Koch (1999) on the subject of homesickness and proposes that the Creative Arts Therapies can be an effective intervention in the acculturation process with refugees.

It is a timely paper which will attract the interest of readers and particularly that of researchers.

One minor change I would like to suggest is that in English one would not refer to something turning pathological but rather becoming pathological. 

Author Response

Dear Reviewer,

Thank you for taking the time to review our manuscript. We have attended to your suggestions and made the indicated changes.

Thanks again!

Round 2

Reviewer 1 Report

Thank you for responding the review report provided. You have made a valuable contribution. Thank you and continue the good work. 

Author Response

Dear Academic Editor,

Thank you for your response to our manuscript.  We have made changes in accordance with your suggestions. Please find them highlighted in blue (as opposed to the red highlights from our earlier revision). To summarize, we have reframed the paper by changing the title to reflect that this is first and foremost a literature review. We have also toned down our emphatic claims throughout the paper as we give only one empirical example.

Best,

Rebekka Dieterich-Hartwell
